# The Relationship between Physical Activity, Physical Exercise, and Human Gut Microbiota in Healthy and Unhealthy Subjects: A Systematic Review

**DOI:** 10.3390/biology11030479

**Published:** 2022-03-21

**Authors:** Stefania Cataldi, Valerio Bonavolontà, Luca Poli, Filipe Manuel Clemente, Michele De Candia, Roberto Carvutto, Ana Filipa Silva, Georgian Badicu, Gianpiero Greco, Francesco Fischetti

**Affiliations:** 1Department of Basic Medical Sciences, Neuroscience and Sense Organs, University of Study of Bari, 70124 Bari, Italy; stefania.cataldi@uniba.it (S.C.); valerio.bonavolonta@uniba.it (V.B.); luca.poli@uniba.it (L.P.); michele.decandia@uniba.it (M.D.C.); roberto.carvutto@uniba.it (R.C.); francesco.fischetti@uniba.it (F.F.); 2Escola Superior Desporto e Lazer, Instituto Politécnico de Viana do Castelo, Rua Escola Industrial e Comercial de Nun’Álvares, 4900-347 Viana do Castelo, Portugal; filipe.clemente5@gmail.com (F.M.C.); anafilsilva@gmail.com (A.F.S.); 3Research Center in Sports Performance, Recreation, Innovation and Technology (SPRINT), 4960-320 Melgaço, Portugal; 4Instituto de Telecomunicações, Delegação da Covilhã, 1049-001 Lisboa, Portugal; 5The Research Centre in Sports Sciences, Health Sciences and Human Development (CIDESD), 5001-801 Vila Real, Portugal; 6Department of Physical Education and Special Motricity, Transilvania University of Brasov, 500068 Brasov, Romania; georgian.badicu@unitbv.ro

**Keywords:** human microbiota, physical activity, physical exercise, training, microbiome, gut

## Abstract

**Simple Summary:**

To date, the influence that physical activity (PA)/physical exercise (PE) can exert on the human gut microbiota (GM) is still poorly understood. Several issues arise in structuring research in this area, starting from the association between PA/PE and diet. Indeed, the diet of an individual is a key factor for the composition of the GM and those who regularly practice PA/PE, generally, have dietary patterns favorable to the creation of an ideal environment for the proliferation of a GM capable of contributing to the host’s health. It is therefore difficult to establish with certainty whether the effects generated on the GM are due to a PA protocol, the type of diet followed, or to both. In addition, most of the available studies use animal models to investigate a possible correlation between PA/PE and changes in the GM, which may be not necessarily applied to humans. Evidence suggests that aerobic PA/PE seems capable of producing significant changes in GM; training parameters, likewise, can differentially influence the GM in young or elderly people and these changes appear to be transient and reversible.

**Abstract:**

Several studies have been conducted to find at least an association between physical activity (PA)/ physical exercise (PE) and the possibility to modulate the gut microbiome (GM). However, the specific effects produced on the human GM by different types of PA/PE, different training modalities, and their age-related effects are not yet fully understood. Therefore, this systematic review aims to evaluate and summarize the current scientific evidence investigating the bi-directional relationship between PA/PE and the human GM, with a specific focus on the different types/variables of PA/PE and age-related effects, in healthy and unhealthy people. A systematic search was conducted across four databases (Web of Science, Medline (PubMed), Google Scholar, and Cochrane Library). Information was extracted using the populations, exposure, intervention, comparison, outcomes (PICOS) format. The Oxford Quality Scoring System Scale, the Risk of Bias in Non-Randomized Studies of Interventions (ROBINS-I) tool, and the JBI Critical Appraisal Checklist for Analytical Cross-Sectional Studies were used as a qualitative measure of the review. The protocol was registered in PROSPERO (code: CRD42022302725). The following data items were extracted: author, year of publication, study design, number and age of participants, type of PA/PE carried out, protocol/workload and diet assessment, duration of intervention, measurement tools used, and main outcomes. Two team authors reviewed 694 abstracts for inclusion and at the end of the screening process, only 76 full texts were analyzed. Lastly, only 25 research articles met the eligibility criteria. The synthesis of these findings suggests that GM diversity is associated with aerobic exercise contrary to resistance training; abundance of *Prevotella* genus seems to be correlated with training duration; no significant change in GM richness and diversity are detected when exercising according to the minimum dose recommended by the World Health Organizations; intense and prolonged PE can induce a higher abundance of pro-inflammatory bacteria; PA does not lead to significant GM α/β-diversity in elderly people (60+ years). The heterogeneity of the training parameters used in the studies, diet control, and different sequencing methods are the main confounders. Thus, this systematic review can provide an in-depth overview of the relationship between PA/PE and the human intestinal microbiota and, at the same time, provide indications from the athletic and health perspective.

## 1. Introduction

It is now well established that several microbial communities coexist with our organism; most of these—including bacteria, archaea, fungi, and even viruses—are hosted in the human gastrointestinal tract and, for this reason, they are generally referred to as the gut microbiota [1]. These microbes have developed a symbiotic relationship with our organism, as they are essential for our survival, and this is due to their ability to express a number of genes approximately 150 times greater than the entire human genome [2], which collectively are referred to as the microbiome.

Several studies [3,4] demonstrate that the gut microbiome is a mediator between many factors such as genetics, diet, exercise, and the external environment. While this interaction has been studied extensively in diet-induced changes in the microbiota [5] and host disease processes [6,7], there has been little focus on other factors such as PA/PE. Chronic diseases—such as obesity, metabolic syndrome, diabetes, atherosclerosis, liver dysfunction, and inflammatory bowel disease (IBD)—have been associated with alterations in the GM [8]. Moreover, increasing evidence shows that the complex bidirectional system of communication between the central and enteric nervous systems—named the gut–brain axis (GBA)—can be influenced by the human microbiota, which in this way acts as an emotional and behavioral modulator [9].

The first studies on the human GM date back to about 40 years ago [10], but this remains a relatively young field of research where there are few certainties and much remains to be investigated. Only recently, however, a possible interaction between PA/PE and the intestinal microbiota has been proposed. Only in the last decade have the first studies appeared reporting that PE is able to enrich the diversity of the human intestinal microflora [11] and consequently, improve the health status of the host. It has also been shown that there is a correlation between cardiorespiratory fitness (CRF) and increased microbial diversity in healthy subjects; thus, an improved ability of the heart, lungs, and muscles to take up and consume oxygen for energy is positively correlated with a more diverse microbial profile that is able to promote butyrate production, both factors associated with overall host health. This has allowed the suggesting of exercise prescriptions as a parallel therapy to control diseases associated with a state of dysbiosis [12].

Furthermore, studies carried out first in animal samples [13], and later in humans [14], have highlighted the ability of PE to modify the composition and functionality of the microbiota independently of diet and body composition, thus in both lean and obese, as well as previously sedentary subjects. Considering, moreover, the mutual influence that is established between the microbiota and the central nervous system, it is possible that PA/PE can somehow exert some effects even on that complex system previously mentioned and known as GBA; this is the conclusion reached by recent work [15] according to which the current evidence suggests that PE is able to mediate this bidirectional relationship between the gut and the brain through the modifications induced at the level, precisely, of the intestinal microbiota. Thus, PA/PE could become a strategy and therapeutic factor for gastrointestinal and psychological problems. Only recently has a new definition, the muscle–gut–brain axis, been introduced, suggesting that exercise-mediated changes at the level of the human microbiota could influence the development and progression of neurodegenerative diseases such as Alzheimer’s that currently have no cure [16].

The mechanisms by which PA/PE may lead to alterations in the human GM are poorly understood. Several hypotheses are currently proposed: changes in the profile of bile acids, which may have an antimicrobial function and exert selective pressure on certain bacterial strains; suppression of TLR4 (Toll-like receptor 4) signaling pathways, which can reduce serum LPS levels; increased production of intestinal immunoglobulin A (IgA), which leads to increased resistance to colonization by specific commensal microbes; reduced intestinal transit time; activation of the hypothalamic–pituitary–adrenal (HPA) axis and subsequent hormone production following physical stress [17]; hormetic effect of stress induced by PA/PE, at the intestinal level, that can stimulate beneficial adaptations of the intestinal barrier [18].

However, this field seems to permeate different aspects of human health in a capillary manner, branching out to the current COVID-19 emergency. In fact, a recent work [19], following the equally recent evidence available [20,21,22], theorizes that intestinal microbiota dysbiosis associated with poor integrity of the intestinal barrier could allow SARS-CoV-2 to easily access the enterocytes circulating and infecting organs expressing the ACE2 receptor, thus increasing the chances of contracting the infection and/or exacerbate the symptoms. If future studies were to empirically demonstrate the connection between the intestinal microbiota and COVID-19, the relevance of an adequate composition of the intestinal flora would be further emphasized, and PA/PE would appear to contribute to this.

Therefore, the present systematic review aims to summarize the evidence on the bi-directional relationship between PA/PE and the human GM, providing more knowledge and details; specifically, the aims were threefold: to investigate the effects induced, on the GM, by different types of PA/PE; to investigate the effects induced, on the GM, by a different training frequency, intensity, and volume; and to investigate the age-related effects of PA on GM and the persistence of induced changes.

## 2. Methods

### 2.1. Data Sources and Search Strategy

This systematic review was carried out following the PRISMA statement [23]. The protocol was registered in PROSPERO (International Prospective Register of Systematic Reviews; code: CRD42022302725). Two team authors performed initial research questions which were then transformed according to the participants, intervention, comparators, outcomes, and study design (PICO_S_) system. The databases used for the identification of scientific articles were Web of Science, Medline (PubMed), Google Scholar, and Cochrane Library. Searches were performed from September to November 2021. Candidate studies were identified by using the following Boolean search syntax: “((microbiota or microbiome) and (exercise or “physical exercise”)”/“(microbiota or microbiome) and (activity or “physical activity”)”/“(microbiota or microbiome) and (sport or athletes))”. Afterward, the following filters were activated—text availability: full text; species: humans; languages: English. The search strategy used for the PubMed database was a combination of the MeSH database and Boolean search syntax. After candidate articles were collected, further identification was conducted based on inclusion and exclusion criteria.

### 2.2. Elegibility Criteria

The inclusion criteria for this systematic review were only English-language original peer-reviewed articles, published from January 2010 to December 2021. Excluded records were review articles, meta-analyses, practical guidelines, unpublished studies, books and book chapters, editorials, letters to the editor, and essays, although they were used as a reference to identify the original search to examine for inclusion. In addition, the framework of population, intervention, comparator, outcomes, and study design (PICOs) was followed to incorporate the studies into a systematic review (Table 1).

### 2.3. Data Extraction and Synthesis

Two team authors independently extracted relevant information from included studies using a Microsoft Excel spreadsheet: author, year of publication, study design, number and age of participants, type of PA/PE carried out, protocol/workload (training volume, frequency, intensity) and diet assessment, duration of intervention, measurement tools used, and outcomes. Any disagreements were resolved by consensus. The characteristics of the studies were summarized, and data on the effects of different types of PA/PE on the metabolic variables of GM were qualitatively synthesized.

### 2.4. Data Items

The following information was extracted from the included articles: changes in microbial α-diversity (differences within one sample) and β-diversity (assessment of differences or similarities in diversity between two samples); relative abundance of specific bacteria (phylum, class, order, family or species); fecal metabolites; *Firmicutes:Bacteroidetes* ratio.

Additionally, the following data items were extracted: number of participants, age-group (young, adult, elderly), sex (man, woman or both), training or physical activity level (untrained, trained, physical active or inactive), type of PA/PE performed (aerobic, anaerobic or both), modalities of diet assessment, type of gut microbiota analysis (16S rRNA amplicon sequencing, metagenomic shotgun sequencing, fecal metabolomics or combination of these); characteristics of experimental settings and procedure, dietary patterns, if available, and their possible influences on the outcomes will be described. We only included GM data extracted from fecal samples, thus data extracted through intestinal biopsy or other biological samples were excluded to avoid heterogeneity in results.

### 2.5. Risk of Bias within Studies

The modified version of the Jadad Scale (Table 2) was used to assess risk of bias within randomized controlled studies. The scale includes five items, total score for each article ranged from 0 to 5 and was computed by summing the score of each item. The Risk of Bias in Non-Randomized Studies of Interventions (ROBINS-I) tool (Table 3) was used to assess the quality of non-randomized studies of interventions (NRSIs). The tool includes seven items and an overall outcome for each study resulting from a detailed evaluation. Lastly, the JBI Critical Appraisal Checklist for Analytical Cross-Sectional Studies (Table 4) was used to assess risk of bias within cross-sectional articles. The checklist includes eight items and an overall appraisal. All analyses were independently assessed for methodological quality by two reviewers.

## 3. Results

### 3.1. Identification of Studies

At the end of the selection process, 1189 articles were extracted, of which *n* = 663 from Web of Science, *n* = 452 from PubMed, *n* = 65 from Google Scholar, and *n* = 9 from Cochrane Library. Each title and abstract were screened for relevance, removing review articles, unpublished studies, meta-analyses, practical guidelines, books and book chapters, editorials, letters to the editor and essays (*n* = 618). Thereafter, the search strategy was based on the assessment of the full text of the remaining 76 articles to verify their eligibility. Lastly, 25 research articles specifically focusing on GM responses to PA/PE, in healthy and unhealthy participants, were included (Figure 1).

### 3.2. Study Characteristics

Thus, this systematic review provides an evaluation of 25 studies [11,12,14,24,25,26,27,28,29,30,31,32,33,34,35,36,37,38,39,40,41,42,43,44,45] inquiring in-depth about the PA/PE-induced changes on the human GM. Most of them (*n* = 11) [12,14,25,26,29,34,36,38,41,42,44] regarded GM modifications following aerobic PA/PE, while other works (*n* = 2) [24,32] focused on changes occurring after an endurance protocol, few studies (*n* = 3) [27,30,37] focus on the changes induced by concurrent training or (*n* = 3) [31,33,35] separate interventions of resistance and aerobic exercises, lastly, further works (*n* = 6) [11,37,39,40,43,45] investigated the effects produced on the GM following the practice of specific sports. Table 5 summarizes the characteristics of the studies included.

### 3.3. Outcome Measures of the Included Studies

The main outcome measures included in this analysis measured GM modifications (76%; *n* = 19) through the use of 16S rRNA amplicon sequencing performed on fecal samples [11,12,14,24,25,26,28,29,30,31,33,34,35,36,37,41,42,44,45]; metagenomic whole-genome shotgun sequencing and fecal metabolomics, in three studies [27,40,43]; metagenomic whole-genome shotgun sequencing and RNA sequencing performed on fecal samples, in one study [39]; 16S rRNA amplicon sequencing and metagenomics, in another one [32]; lastly, one study used 16S rRNA hybridization and DNA-staining [38].

## 4. Discussion

### 4.1. Novelty of This Review

A small number of systematic reviews have attempted to investigate and summarize the effects of PA or PE on the human GM, in particular a recent work by Aya et al. [46] focuses on this topic, considering healthy subjects aged 18–45 years. Our review also looks at studies that consider subjects with certain diseases so that the effects of PA/PE on the human GM from the perspective of the unhealthy population can be assessed as well. In addition, we analyzed the most recent studies [29,30,31,35,45] not evaluated in the previous reviews, which can consolidate some of the aspects that have previously emerged.

We have focused our analysis not only on the effects of PA/PE on the GM, but we also explored how certain training parameters (intensity, frequency, and volume) can differently affect the human GM.

Furthermore, when analyzing the results, we separated those obtained from groups of athletes and those from the general population from those relative to elderly subjects (60+ years) to investigate the age-related effects of PA/PE on GM and the persistence of induced changes.

### 4.2. Different Types of PA/PE and Their Influence on GM

Most of the studies carried out on humans consider aerobic physical activity to assess the changes induced in the GM. Three studies [27,28,30] administered a mixed intervention protocol based on resistance training (RTE) plus cardiorespiratory exercise (CRE) or strength training plus endurance training, three studies [31,33,35] proposed separate interventions (RTE or CRE) with the same research design.

Another study reported that GM diversity in young and healthy subjects is associated with aerobic fitness, furthermore, GM adaptation to the demands of increased physical fitness would appear to be linked to several functional cores rather than specific bacterial groups. Moreover, the microbial profile of subjects with high levels of cardiorespiratory fitness (CRF) appears to stimulate the production of butyrate, probably through an increase in genera such as *Clostridiales*, *Roseburia*, *Lachnospiraceae*, and *Erysipelotrichaceae* [12]. Subjects with low aerobic fitness show a higher presence of *Eubacterium rectale-Clostridium coccoides* (EreC), associated with obesity and related metabolic disorders [47]; *Enterobacteria*; and a lower presence of *Bacteroides*. However, results seem to be confounded by adiposity, since the relationship between maximal aerobic power (VO_2max_) and EreC disappeared after adjusting for body fat% [38]. There appears to be also a relationship between VO_2max_ and the *Firmicutes*:*Bacteroidetes* ratio [41], supporting what has been observed in previous studies [12,14,38] and suggesting that PE can induce favorable changes in the composition of the GM in healthy subjects. Endurance activity, in line with what has already been observed [14], is able to modestly modify the GM compositionally and functionally without significantly affecting body composition in overweight, previously sedentary subjects [32]. Exercise increased the abundance of the phylum *Verrucomirobia*, and within it the genus *Akkermansia*, associated with beneficial host effects due to its ability to enhance lipid oxidation [48]. Similar changes have been noted in other works [11,36], but here the increase in the genus *Akkermansia* occurred independently of weight, body composition, and diet, suggesting that these taxa are indeed responsive to exercise. Similarly, a reduction in the phylum *Proteobacteria* and an unidentified genus of *Enterobacteriaceae* associated with negative effects on host health was observed [49,50].

One study compares RTE and CRE activity [35], unlike in a previous study where RTE activity was marginal and not isolated [28], observing that cardiovascular, aerobic activity, determines a change within the GM. In the RTE group, on the other hand, no changes in the microbial composition were observed, suggesting that changes in the GM are not a foregone conclusion of exercise but could depend on the way in which the activity is practiced. It is possible, anyway, that no difference was observed due to the lower adherence to microbe sampling, in the RTE group. Only one short-term (six-week) randomized controlled trial [31] divided participants into three groups: endurance, strength, and control. The 16rRNA gene amplicon sequencing did not detect specific bacterial changes in any of the experimental groups, but only a wide variability in changes to the GM of individuals within the same group, suggesting that these modifications may be individual specific. Stool samples of elite athletes were also analyzed and compared to control group subjects who did not exercise. No significant difference was found in α- and β-diversity, contrary to previous studies [11,39]; whereas in line with these observations a higher abundance of different *Bacteroidetes* species was found in the elite athletes, but the lack of dietary data could be a confounder. The main limitation of this study is the control group that showed similar daily step values to the endurance group and even higher values to the strength group, which means that they cannot be considered sedentary subjects, as they performed low-intensity physical activity.

Few studies investigate the influence of exercise on the GM composition of athletes performing endurance activities. A specific correlation between certain taxa, detected only in professional cyclists and high endurance workload was identified, suggesting a possible connection between exercise intensity and specific GM changes. There is a significant correlation between the increased abundance of the genus *Prevotella* and the total duration in hours of weekly training performed (>11 h/week); consistent with previous observations [11] a reduced presence of *Bacteroides* and a relative abundance of the genus *Akkermansia* is observed. A high percentage of *M. smithii* has been also detected in some professional cyclists [39], with an overproduction of genes involved in methane production, which would allow the microbial community to be more energy-efficient [51,52]. Assessing the variation in the GM and metabolome across athletes of different sports, based on static and dynamic components, O’Donovan et al. [43] found the existence of individual variability among athletes with clustering of majority samples driven by the relative abundances of five species, namely *F. prausnitzii*, as previously found by Bressa et al. [36]; *E. rectale*; *P. necessaries*; *B. vulgatus*; and *G. massiliensis*. Variability, possibly due to variance in duration and exercises modalities, was observed in absence of significant differences in dietary intakes across athletes from different sports. After a single bout of moderate-intense exercise, an increase in the tryptophan, tyrosine, and phenylalanine metabolites—essential amino acids (EAAs) synthesized by GM—was observed in 40 male endurance cross-country athletes, and in particular a strong cross-talk between GM and systemic tryptophan metabolism highlighted by changes in the abundance of *E. coli* and several bacterial taxa, such as *Romboutsia*, *Ruminococcocaceae* UCG-005, *Blautia*, *Ruminiclostridium* 9, and *Clostridium phoceensis* [45], which possess tryptophan synthesis machinery [53].

### 4.3. Gut Microbiota and Different Physical Activity Levels

Several cross-sectional studies tried to evaluate the specific relationship between PA/PE and GM, one of the earliest studies done in humans [11] was conducted on professional rugby players during preseason training. Results were correlated to diet, since it was not possible to isolate PE from the diet of professional athletes. A higher richness and α-diversity were observed in the athletes compared to the control groups, which was also detected later by another study [37]. Abundance in the group of athletes was observed in some specific groups while at the phylum level the highest diversity was found in *Firmicutes*, particularly in the genus *Faecalibacterium prausnitzii*, producers of butyrate [54,55] and markers of intestinal health [56]. Furthermore, abundance in the *Akkermansiaceae* family and genus *Akkermansia* has been found in athletes and the low BMI control group, which appears to be inversely correlated with obesity and associated metabolic disorders [48]. Lack of dietary and other external controls are the main confounders. Sequencing (shotgun) of the metagenome in its entirety to investigate the taxonomic composition and functional potential [40] has shown that the differences in the GM of athletes compared to the sedentary group is even more pronounced at a functional level than merely compositional. Contrary to the observations of Clarke et al. [11], comparing the composition of the intestinal microbiota of subjects who do not practice any type of PA and subjects who exercise at the minimum dose recommended by the World Health Organization (WHO) for adults aged 18–64 years [57], it seems that PA does not lead to significant changes in microbial richness and diversity (α and β). As a possible cause of this discrepancy, different exercise modalities and intensities are indicated [36]. A similar protocol was implemented through a randomized controlled trial with a 10-week moderate-intensity aerobic intervention [29]. Similarly, no significant difference was found in bacterial diversity (α and β), as already observed in other studies [14,32], nor at phylum, class, order, family, or species level. The same conclusion was reached by the only study linking short-term high-intensity interval training (HIIT) and changes in the GM [34]. This suggests that changes in diversity (α and β) require a greater and/or longer stimulus.

Since hormonal set-up can affect GM composition [58], 40 premenopausal women were divided into two groups, active (ACT) and sedentary (SED). No changes are present in the *Bacteroidetes:Firmicutes* ratio, but consistent with what has been previously reported [11], a downward trend is observed regarding the presence of *Bacteroidetes* in the ACT group. In agreement with preclinical studies [59,60,61] in the ACT group, a lower proportion of the family *Turicibacteraceae* was observed, as well as a significant abundance of some species such as *R. hominis*, *A. muciniphila*, and *F. prausnitzii*—species associated with positive effects on host health [62,63,64]. A greater presence of the *Coprococcus* genus has been found in ACT subjects, the scarcity of which has been associated with IBD [65]. The amount of fiber intake was significantly higher in the ACT group than in the SED group which, in turn, consumed more processed meats, all of which may play a key role in shaping the gut microbiota [66].

Topological analysis of the microbial network [44] suggests that the transition from an active to a sedentary lifestyle leads to changes in those bacterial taxa considered ‘key’ to host health and, in the opposite case to a modification of those ‘key’ bacteria associated with diseases. Among the most relevant in the reorganization that occurs during the transition from ACT to SED are the species *Roseburia faecis* and an unclassified species of *Roseburia* which are considered markers of good host health [64,67]. Within the bacterial network of ACT subjects, the *Rikenellaceae* and *Erysipelotrichaceae* families have been detected, whose role in humans, however, is still not entirely clear [11,68]. Finally, playing a key role in the transition from SED to ACT would appear to be some unclassified species of the genus *Sutterella*, recently associated with neurological disorders [69], conditions such as ulcerative colitis with impaired immune system function [70], as well as being identified as a typical driver in diabetes [71]. Strong limitations remain represented by the different dietary habits between ACT and SED, the former consumed more dietary fibers while the latter consumed greater amounts of processed meats and sugars.

Athletes who engage in intense and prolonged PE show a particular composition of the microbiota, characterized by a higher abundance of bacteria involved in inflammatory processes, such as *Haemophilus* and *Rothia* genus [72], *Mucispirillum* [73,74], and *Ruminococcus gnavus* [75]. Interestingly, a relative abundance of *Faecalibacterium*, a butyrate-producing bacterium usually known to be beneficial for human’s health [76,77,78], has been observed concomitantly with a high abundance of pro-inflammatory bacteria in female endurance runners whose abnormal gut environment can lead it to act like an opportunistic bacterium [79]. Excessive PA/PE leads to stress that generates intestinal permeability [80], which results in the release of bacteria and products, such as lipopolysaccharides (LPS), recognized as toxic by the body, which triggers an inflammatory response [81,82].

In both high-intensity and moderate-intensity PA, there is a reduction in intestinal inflammation with concomitant changes in the microbial profile [26]. A reduction in the *Firmicutes:Bacteroidetes* ratio is noted, with an increase in *Bacteroidetes*—which could be beneficial for athletes by playing a key role in the metabolic conversion of complex sugar polymers and protein degradation [83,84,85,86]—and a reduction in the genus *Clostridium* and *Blautia*, which would appear to play a key role in the immune response [87,88,89]. In a recent randomized controlled trial [25], using a specific training protocol called ACTIWE [90], changes in β-diversity were observed in both subjects performing intensive (VIG) or moderate-intensity (MOD) physical activity and a reduction in heterogeneity only in the VIG group. Changes in α-diversity were also observed in the MOD group, in contrast to Munukka et al. [32], whereas the VIG group experienced the greatest increase at 3 months post-intervention, compared to the control group, suggesting that more intense activity may be required to induce changes in the GM of previously sedentary overweight and obese subjects.

High intensity physical activity seems to be capable of inducing alterations in the GM correlated closely with improvements in glucose homeostasis and insulin sensitivity regardless of body weight and fat mass reduction [27]. Increased gene expression of functional pathways inducing the production of SCFAs and degradation of BCAAs is observed, which may be related to improvements in glucose metabolism as increased BCAA concentrations have been associated with insulin resistance [91,92,93,94].

### 4.4. Age-Related Changes on GM Induced by PA and How Long They Persist

As it has been observed that the changes produced by PA on the microbiome are not unidirectional, as multiple factors can influence the outcome, including the host’s age [95,96,97]. Furthermore, when compared to young adult populations, elderly subjects show a reduction in the diversity of the microbiota, which is usually characterized by a large interindividual variability [98]. An increase in the relative abundance of the genus *Bacteroides* in parallel with an improvement in CRF has been detected [33], as a result of 12-week performing brisk walking in healthy elderly women (65+ years), supporting the observation made by Yang et al. [38] of an association between CRF and the presence of *Bacteroides*. Generally, bacterial species belonging to the genus *Bacteroides* can contribute to a reduction in metabolic dysfunction [99]. The narrowness of the sample used, the absence of previous habits in the performance of PA, and the higher energy expenditure induced by aerobic exercise may represent some limitations. Similarly [24], after 5 weeks of aerobic exercise, in agreement with Bressa et al. [36], no changes in microbial diversity (α and β) were observed in elderly men (60+ years), only minor changes such as increased relative abundance of the genus *Oscillospira*, positively associated with leanness [100,101], and reduced *C.difficile* suggesting a beneficial effect of PA operating through toxin reduction produced by this bacterium. The results of this study are, however, limited by its short duration, lack of dietary control, and the exclusive use of healthy elderly male subjects.

No significant changes in α-diversity were detected in previously sedentary elderly women after 8 weeks of aerobic and resistance exercise. A partial change in the relative abundance and structure of the GM was observed, such as an increase in *Betaproteobacteria*, *Burkholderiales,* and *Prevotella* [30], as previously noted by Petersen et al. [39]. A modest association was found between PA levels and specific gut microbes among community-dwelling elderly men. PA was not associated with α-diversity and just a slight association with β-diversity was observed [42]. Comparing lifetime elderly endurance athletes (LA) with healthy controls (CTRL) who met the American College of Sports Medicine (ACSM) recommendations on physical activity for older adults [102], no significant difference was observed in the GM diversity [103], a lower *Bacteroidete:Prevotella* ratio was detected in the LA group, which could be a predictive marker of weight and fat loss [104].

To date, it is not entirely clear how long it takes for physical activity-induced changes in the GM to become stable. Aerobic activity appears to induce changes in the GM, but these changes did not seem to persist during, or until the end, of a short-term exercise protocol (8 weeks); in fact, these changes were more pronounced at the beginning of the program, subsequently decreasing until they became irrelevant [35]. This suggests that persistent microbiome differences between athletes and non-athletes [11,37,40] may be the result of long-term lifestyle differences between these groups that cannot be quickly achieved.

### 4.5. Limitations and Future Recommendations

Given the many factors that can interact with and modify human GM (e.g., type of birth, genetics, nutrition, age, antibiotics, stress), extrapolating the effects induced directly by PA/PE becomes a complex process. In particular, the type of diet, when not controlled, is a major confounding element; but even when attempts are made to track and standardize it, margins of uncertainty persist. In addition, the different exercise modalities—frequency, intensity, and volume used by the researchers—can contribute to the discrepancies in the results. Finally, different gene sequencing methods (16S rRNA, ITS rRNA, Shotgun metagenomic) can lead to different results under similar experimental conditions, because of their different sensitivities in microbial identification. In the currently available studies, therefore, these critical aspects lead to a very cautious consideration of the results obtained.

Thus, further randomized controlled trials are needed, with even more scientific rigor and greater focus on aspects that are often overlooked—such as the presence of viruses and bacteriophages in the gut and metabolomic analysis—controlling as much as possible for the various confounding factors that can substantially influence the results.

## 5. Conclusions

PA/PE appears to be able to counteract changes in GM related to obesity and T2D; as well as being able to reduce inflammatory signals by creating a controlled environment for GM. In addition, the ability to stimulate bacterial community richness has been observed. Furthermore, the GM of athletes may have enhanced functional capabilities such as tissue repair and increased energy gain from carbohydrate metabolism.

From a practical point of view, some indications may arise as follows: higher levels of CRF result in greater bacterial diversity, regardless of diet; aerobic activities appear to be able to induce significant changes in the composition of the GM, in contrast to resistance activities, even if GM modifications could be individual-specific; the training modalities and intensity can affect the GM differently; volume and frequency recommended by the WHO (the minimum dose of physical activity) for adults aged 18–64 years appears to prompt some changes in the composition of the GM, but not significant changes in terms of richness and diversity; a medium–low intensity training protocol produces limited changes in the GM, higher intensity seems to be necessary to induce changes in overweight and obese, previously sedentary subjects; strenuous and/or excessively prolonged exercise may negatively impact the GM. However, the ideal dose–response relationship of exercise remains uncertain, given the multiplicity of exercises and their dynamics of intensity and volume, which have not yet been fully studied. Factors that can produce adaptive changes in the structure of the GM, including PA/PE, may have a greater and more lasting impact if exploited early in life; exercise-induced changes are transient and most likely dependent on repeated stimuli over time.

## Figures and Tables

**Figure 1 biology-11-00479-f001:**
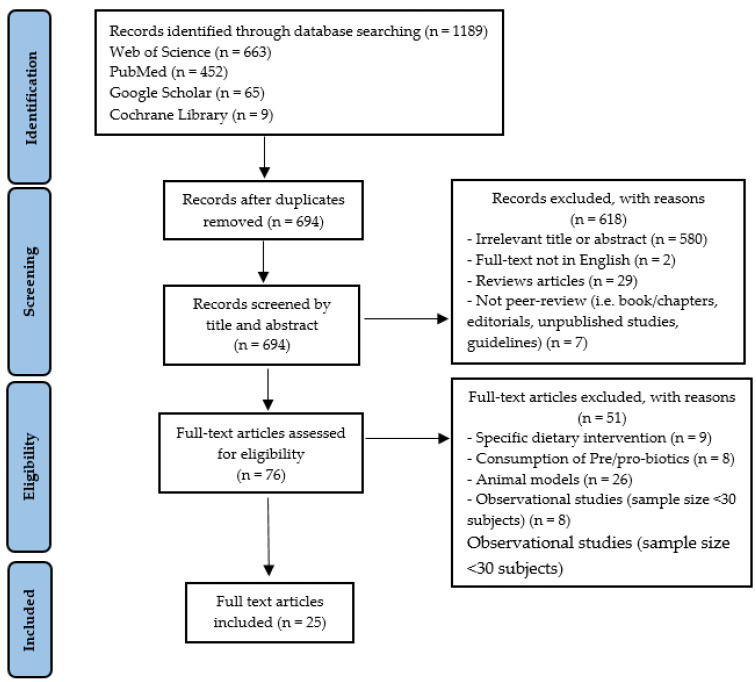
Study selection and eligibility screening flow according to PRISMA guidelines.

**Table 1 biology-11-00479-t001:** Selection criteria used in the systematic review.

	Inclusion Criteria	Exclusion Criteria
Population	Healthy and unhealthy subjects, no age restrictions, both sexes (from sedentary to athlete subjects).	Subjects who take or have taken (in the month before the intervention) pre/pro-biotics and/or antibiotics.
Intervention/Exposure	Intervention with any kind of PE protocol or PA/PE exposure.	Intervention with a specific dietary protocol.
Comparator	Intervention that has a control group running a different PA/PE protocol or none, a comparison subgroup, or at least a pre/post-intervention comparison.	Absence of any kind of control/comparison.
Outcome(s)	Measures of differences for α and β diversity, relative abundance of specific bacteria, metabolomic and metagenomic data analyzed with any kind of sequencing tool.	Lack of baseline and/or follow-up data, or absence of at least one of the measurements indicated in the inclusion criteria—outcome(s).
Study Design	Studies with experimental design (randomized and non-randomized trial), observational studies (sample size >30 subjects).	Observational studies (sample size *<* 30 subjects), case study.

**Table 2 biology-11-00479-t002:** The modified Oxford quality scale.

Authors	Was the Treatment Randomly Allocated?	Was the Randomization Procedure Described and Was it Appropriate?	Was There a Description of Withdrawals and Dropout?	Was There a Clear Description of the Inclusion/Exclusion Criteria?	Were the Methods of Statistical Analysis Described?	Jadad Score (0–5)
Taniguchi et al. (2018) [24]	Yes	Yes	Yes	Yes	Yes	5
Kern et al. (2020) [25]	Yes	Yes	No	Yes	Yes	4
Motiani et al. (2020) [26]	Yes	Yes	Yes	Yes	Yes	5
Liu et al. (2020) [27]	Yes	Yes	No	Yes	Yes	4
Quiroga et al. (2020) [28]	Yes	Yes	Yes	Yes	Yes	5
Resende et al. (2021) [29]	Yes	Yes	Yes	Yes	Yes	5
Zhong et al. (2021) [30]	Yes	Yes	Yes	Yes	Yes	5
Moitnho-Silva et al. (2021) [31]	Yes	Yes	Yes	Yes	Yes	5

**Table 3 biology-11-00479-t003:** ROBINS-I.

Study	Bias Due to Confounding	Bias in Selection of Participants into the Study	Bias in Classification/ Measurement of Intervention	Bias Due to Deviations from Intended Interventions	Bias Because of Missing Data	Bias in Measurement of Outcomes	Bias in Selection of the Reported Result	Overall
Allen et al., 2018 [14]	Low	Moderate	Moderate	Moderate	Moderate	Low	Low	Moderate
Munukka et al., 2018 [32]	Low	Moderate	Low	Moderate	Moderate	Moderate	Moderate	Moderate
Morita et al., 2019 [33]	Low	Low	Low	Moderate	Moderate	Moderate	Low	Moderate
Rettedal et al., 2020 [34]	Low	Low	Low	Low	Low	Low	Low	Low
Bycura et al., 2021 [35]	Low	Low	Low	Moderate	Moderate	Low	Low	Moderate

**Table 4 biology-11-00479-t004:** JBI Critical Appraisal Checklist for Analytical Cross-Sectional Studies.

Study	Where the Criteria for Inclusion in the Sample Clearly Defined?	Where the Study Subjects and the Setting Described in Detail?	Was the Exposure Measured in a Valid and Reliable Way?	Were Objective, Standard Criteria Used for Measurement of the Condition?	Were Confounding Factors Identified?	Were Strategies to Deal with Confounding Factors Stated?	Were the Outcomes Measured in a Valid and Reliable Way?	Was Appropriate Statistical Analysis Used?	Overall Appraisal (Included/Excluded)
Clarke et al. (2014) [11]	Y	Y	Y	Y	Y	Y	Y	Y	I
Estaki et al. (2016) [12]	Y	Y	Y	Y	Y	Y	Y	Y	I
Bressa et al. (2017) [36]	Y	Y	Y	Y	Y	Y	Y	Y	I
Mörkl et al. (2017) [37]	Y	Y	Y	Y	Y	N	Y	Y	I
Yang et al. (2017) [38]	Y	Y	Y	Y	Y	N	Y	Y	I
Petersen et al. (2017) [39]	Y	Y	Y	Y	Y	N	Y	Y	I
Barton et al. (2018) [40]	Y	Y	Y	Y	N	N	Y	Y	I
Durk et al. (2019) [41]	Y	Y	Y	Y	N	N	Y	Y	I
Langsemo et al. (2019) [42]	Y	Y	Y	Y	Y	N	Y	Y	I
O’Donovan et al. (2020) [43]	Y	Y	Y	Y	N	N	Y	Y	I
Castellanos et al. (2020) [44]	Y	Y	Y	Y	N	N	Y	Y	I
Tabone et al. (2021) [45]	Y	Y	Y	Y	Y	Y	Y	Y	I

Y = yes; N = no; I = included; E = excluded.

**Table 5 biology-11-00479-t005:** Summary characteristics of reviewed studies.

Authors	Study Design	Sample	Subjects Age (years)	Type PA/PE	Protocol/ Workload Assessment	Diet Assessment	Duration Intervention	GM Analysis System	Main Outcomes
Clarke et al., 2014 [11]	Cross-sectional	*n* = 86 (M) elite professional rugby players (*n* = 40) (BMI 29.1 ± 2.9), healthy control (*n* = 46) (23: BMI ≤ 25—23: BMI > 28)	Elite: 29 (±4) Control: 29 (±6)	Rugby PE: aerobic-anaerobic	/	187-food items FFQ. Macronutrients, fiber, and supplement intake	/	16S rRNA GA V4 region	Athletes: ↑ α-diversity, ↑ diversity Firmicutes (phylum), ↑ Prevotella, ↓ Bacteroides, ↓ LactobacillusAthletes/Low BMI: ↑ Akkermansia (genus)
Estaki et al., 2016 [12]	Cross-sectional	*n* = 39 (M/F) healthy subjects, stratified by CRF(Low; Average; High)	L: 25.5 (±3.3) A: 24.3 (±3.7) H: 26.2 (±5.5)	PE: aerobic	/	24 h dietary recall interview. Macronutrients, fiber, saturated fat, and PUFA intake	/	16S rRNA GA V3/V4 region	VO_2_peak positively associated with ↑ GM diversity; ↑ CRF = ↑ taxa producers SCFAs. No differences in α and β-diversity
Bressa et al. 2017 [36]	Cross-sectional	*n* = 40 (F) active (ACT) (*n* = 19) and sedentary (SED) (*n* = 21) subjects, defined by WHO recommendations	ACT: 30.7(±5.9) SED: 32.2(±8.7)	PE: aerobic	/	97-food items FFQ. Macronutrients, fiber, and main food intake	/	16S rRNA GA V3/V4 region	ACT: PA ↑ health-promoting bacteria (F.prausnitzii, R.hominis, A.muciniphila) SED: ↑ Barnesiellaceae, ↑ Turicibacter, ↓ Cropococcus No differences in α/β-diversity and at phylum level between groups.
Mörkl et al., 2017 [37]	Cross-sectional	*n* = 106 (F) Anorexia nervosa (AN) patients (*n* = 18), normal weight (NW) (*n* = 26), overweight (OW) (*n* = 22), obese (O) (*n* = 20) and athletes (AT) (*n* = 20)	24.5 (±4.6)	PE: ball sports PE: aerobic-anaerobic	/	Two 24 h recalls. Macronutrients, fiber, Vit D, and magnesium intake	/	16S rRNA GA V1/V2 region	↓ GM α-diversity in obese and AN groups compared to athletes.
Yang et al., 2017 [38]	Cross-sectional	*n* = 71 (F) premenopausal with low (L), moderate (M), high (H) CRF	L: 40.4 (36.9–44.0) M: 39.7 (35.5–43.8) H: 30.6 (25.6–35.6)	PA: aerobic	/	3-days food records (2 weekdays, 1 weekend day). Macronutrients and total energy intake.	/	16S rRNA hybridization and DNA-staining	↓ Bacteroides and ↑ Eubacterium rectale-clostridium coccoides in Low VO_2max_ compared to High VO_2max_ group.
Petersen et al., 2017 [39]	Cross-sectional	*n* = 33 (M/F) professional (*n* = 22) and amateur (*n* = 11) level competitive cyclists	19–49 (Median age 33)	Cycling PE: aerobic-anaerobic	/	Food questionnaire. Macronutrients and alcohol intake.	/	Metagenomic whole-genome shotgun sequencing and RNA sequencing	No significant correlations between taxonomic cluster and professional or amateur level. ↑ Prevotella relative abundance in cyclists training >11 h/week
Barton et al., 2018 [40]	Cross-sectional	*n* = 86 (M) elite professional athletes (*n* = 40), healthy control (*n* = 46) (22: BMI ≤ 25.2—24: BMI ≥ 26.5)	Elite: 29 (±4) Control: 29 (±6)	Rugby PE: aerobic-anaerobic	/	187-food items FFQ. Macronutrients and total energy intake.	/	Genome shotgun sequencing, fecal metabolomics	↑ Pathways (↑ AA biosynthesis, ↑ carbohydrate metabolism) and ↑ fecal metabolites (microbial produced SCFAs) in athletes
Allen et al., 2018 [14]	Longitudinal design	*n* = 32 (M/F) previously sedentary subjects, lean (*n* = 18) and obese (*n* = 14)	Lean: 25.1 (±6.52) Obese: 31.14 (±8.57)	PE: aerobic	30′ to 60′ 3×wk moderate-to-vigorous intensity (60–75% HRR) exercises	7-days dietary records, 3-days food menu before each fecal collection. Macronutrient, micronutrient, and total energy intake	6 weeks	16S rRNA GA V4 region	No β-diversity differences among groups. ↑ SCFAs producing taxa related to BMI (Faecalibacterium: ↑ lean ↓ obese, Bacteroides: ↓ lean ↑ obese). Changes largely reversed after 6wk of inactivity.
Munukka et al., 2018 [32]	Non-randomized trial	*n* = 17 (F) sedentary subjects BMI > 27.5 kg/m^2^	36.8 (±3.9)	PE: endurance	40′ to 60′ 3×wk exercises, low to moderate intensity	3-days food records (2 weekdays and 1 weekend day). Macronutrients, fiber, and total energy intake	6 weeks	16S rRNA GA V4 region and metagenomics.	↑ Akkermansia and ↓ Proteobacteria (exercise-responsive taxa). Changes in GM do not affect systemic metabolites. No differences in α-diversity, slight ↑ β-diversity
Taniguchi et al., 2018 [24]	Randomized crossover trial	*n* = 33 (M) elderly Japanese subjects	62–76	PE: endurance	3xwk ce, 30′ (wk 1/2)—45′ (wk 3/5), with incremental intensity	Self-administered FFQ, semi-weighted 16-days dietary records. Macronutrients and total energy intake.	5 weeks	16S rRNA GA V3/V4 region	No differences in α and β-diversity. ↓ C.difficile, ↑ Oscillospira. Minor changes in GM associated with cardiometabolic risk factors.
Durk et al., 2019 [38]	Cross-sectional	*n* = 37 (M/F) healthy subjects	25.7 (±2.2)	PE: aerobic	/	Instructed to follow their normal diet for 7-days and MyFitnessPal app tracking.Macronutrients, fiber, coffee, alcohol, and total energy intake.	/	16S rRNA GA	VO_2max_ positively associated to ↑ Firmicutes:Bacteroidetes ratio. No differences in α and β-diversity.
Langsetmo et al., 2019 [42]	Cross-sectional	*n* = 373 (M) community-dwelling older subjects	84.0 (3.9)	PA: aerobic	/	Not controlled or recorded	/	16S rRNA GA V4 region	PA not associated with α-diversity, slight association with β-diversity. ↑ Cetobacterium and ↓ Coprobacillus, Adlercreutzia, Eryspelotrichaceae CC-115 in higher step counts subjects.
Morita et al., 2019 [33]	Non-randomized comparative trial	*n* = 32 (F) healthy sedentary elderly subjects, trunk muscle (TM) (*n* = 14) and aerobic exercise (AE) (*n* = 18) intervention	70 (66–75)	PE: aerobic or anaerobic	TM: 1 h weekly resistance training AE: 1 h daily brisk walking ≥3 METs	138-food and beverage items FFQ. Macronutrients, fiber, saturated fat and total energy intake.	12 weeks	16S rRNA GA	↑ Bacteroides relative abundance only in the AE group.
Kern et al., 2020 [25]	Randomized controlled trial	*n* = 88 (M/F) overweight/obese subjects, moderate intensity (*n* = 31) (MOD), vigorous intensity (*n* = 24) (VIG), bicycling (*n* = 18) (BIKE), control (*n* = 14) (CON)	36 (30;41) Median (25th percentile; 75th percentile)	PE: aerobic	MOD: 5×wk LTPA at 50% VO_2peak_ VIG: 5×wk LTPA at 70% VO_2peak_ BIKE: 5×wk active bicycle commuting to and from work (F: 9–15 km/M: 11–17 km daily), self-selected intensity	Food registrations (3 weekdays—1 weekend day), participants were asked to weigh and register intake of food and beverages. Macronutrients, fiber, and total energy itnake.	6 months	16S rRNA GA V4 region	β-diversity changed in all groups compared to CON, ↑ α-diversity in VIG compared to CON. Decreased heterogeneity in VIG. No genera changed significantly.
O’ Donovan et al., 2020 [43]	Cross-sectional	*n* = 37 (M/F) elite athletes from 16 different sports stratified by dynamic and/or static components	27 (±5)	PE: different sports PE: aerobic-anaerobic	/	FFQ. Macronutrients, fiber, beverage, and total energy intake.	/	Metagenomic whole-genome shotgun sequencing and urine and fecal metabolomics	Individual variability among athletes, majority samples driven by 5 species (*E. rectale*, *P. necessaries*, *F. prausnitzii*, *B. vulgatus*, *G. massiliensis*). High dynamic component: most compositionally distinct GM. High dynamic+static components: most functionally distinct GM.
Motiani et al., 2020 [26]	Randomized controlled trial	*n* = 26 (M/F) obese sedentary prediabetic/T2D, sprint interval training (*n* = 13) (SIT), moderate-intensity continuous training (*n* = 13) (MICT)	49 (±4)	PE: aerobic	SIT: 3×wk HIIT 30″ exercise bouts (4-6) cycling (wingate protocol) 4′ recovery between bouts MICT: 40′–60′ 3×wk moderate intensity (60% VO_2peak_) cycling	Not controlled or recorded. Instructed to maintain their dietary habit.	2 weeks	16S rRNA GA V3/V4 region	↑ Bacteroidetes ↓ Firmicutes:Bacteroidetes ratio, ↓ Clostridium and Blautia genus.
Catellanos et al., 2020 [44]	Cross-sectional	*n* = 109 (M/F) healthy subjects, active (*n* = 64) (ACT) and sedentary (*n* = 45) (SED), described by WHO recommendations	ACT: 32.17 (±7.40) SED: 33.69 (±7.96)	PE: aerobic	/	93-food items FFQ. Macronutrients, fiber, ethanol, and total energy intake.	/	16S rRNA GA V3/V4 region	GM network of active people has higher efficiency and transmissibility rate. Key bacteria reorganization from ACT to SED: *Roseburia faecis*, unclassified roseburia spp. Key bacteria reorganization from SED to ACT: unclassified *Sutterella* spp.
Liu et al., 2020 [27]	Randomized controlled trial	*n* = 39 (M) medication naïve overweight/obese pre-diabetic subjects	Responders: 43.29 (±3.27) Non-responders:(36.00 ± 4.55)	PE: aerobic and anaerobic	70′ 3×wk high intensity combined aerobic and resistance interval training, 80–95% HR_max_	FFQ. Macronutrients, fiber, and total energy intake.	12 weeks	Metagenomic whole genome shotgun sequencing and fecal metabolomics	Exercises-induced alterations in the GM correlated with improvement in glucose homeostasis and insulin sensitivity. GM responders: ↑ biosynthesis SCFAs, ↑ BCAA catabolism, ↓ Bacteroides, ↑ Streptococcus mitisGM non-responders: ↑ production of detrimental compounds. No differences in α and β-diversity. Functional capacity of GM can be altered without major shifts in its community structure.
Quiroga et al., 2020 [28]	Randomized controlled trial	*n* = 39 obese pediatric children (*n* = 25) and healthy control (*n* = 14)	7–12	PE: Endurance plus strength	2×wk combined endurance (sprint of 30” max cadence at 3′30″, 4′30″, 5′30″, and 6′30″) and strength training (30–70% 1RM)	Nutritional advice for a healthy and balanced diet.	12 weeks	16S rRNA GA V3/V4 region	↓ Proteobacteria phylum and Gammaproteobacteria class, ↑ Blautia, Dialister and Roseburia genera lead to a GM profile like that of healthy children.
Rettedal et al., 2020 [34]	Non-randomized trial	*n* = 29 (M) overweight (*n* = 15) and lean (*n* = 14) subjects	Overweight: 31 (±2) Lean: 29 (±2)	PE: aerobic	3×wk ce HIIT, 60″ cycling intervals at VO_2peak_ workload interspersed with 75″ rest, 8 to 12 intervals	FFQ for baseline intake. Instructed to maintain normal dietary pattern. Macronutrients, fiber, saturated fat, PUFA, and total energy intake.	3 weeks	16S rRNA GA V3/V4 region	No differences in α and β-diversity. Significant association between the abundance of bacterial spp. (Coprococcus_3, Blautia, Lachnospiraceae_ge, Dorea) and insulin sensitivity marker in the overweight group.
Bycura et al., 2021 [35]	Non-randomized trial	*n* = 56 (M/F) healthy students, cardiorespiratory exercise (*n* = 28) (CRE), resistance exercise (*n* = 28) (RTE)	CRE: 20.54 (1.93) RTE: 21.28 (3.85)	PE: aerobic or anaerobic	CRE: 1 h, 3×wk (2-days group cycling, 1-day rotating CRE activity) 60–90% HR_max_ RTE: 1 h 3×wk full/lower/upper body at 70–85% 1RM	Not controlled or recorded. Instructed to maintain their typical dietary practice and report major deviations.	8 weeks	16S rRNA GA V4 region	CRE: initial changes to GM (wk 2,3) not sustained through or after the intervention. RTE: no changes in microbiome composition.
Resende et al., 2021 [29]	Randomized controlled trial	*n* = 22 (M) healthy previously sedentary subjects, exercise (*n* = 12) and control (*n* = 12)	Exercise: 25.58 (±5.07) Control: 25.5 (±4.66)	PE: aerobic	50′ 3×wk ce at steady speed 60 rpm (wk 1,2), 60/65% VO_2peak_ (wk 3–10, weekly progressiveoverload)	Wk 1: food records Wk 5: 3-day food records Wk 12: 48 h diet record before data collection. Macronutrients, fiber, cholesterol, water, and total energy intake.	10 weeks	16S rRNA GA V4 region	No differences in α- and β-diversity. No significant changes at phylum, class, order, family, or species level. VO_2peak_ positively associated with α-diversity and to the relative abundance of Roseburia, Odoribacter, and Sutterella. BMI positively associated with Desulfovibrio and Faecalibacterium genera.
Tabone et al., 2021 [43]	Cross-sectional	*n* = 40 (M) endurance cross-country runners	35.79 (±8.01)	PE: endurance	/	FFQ, 24 h dietary recall (2 weekdays, 1 weekend day). Macronutrients and total energy intake.	/	16S rRNA GA V3/V4 region	85 serum and 12 fecal metabolites and 6 bacterial taxa (Romboutsia, Escherichia coli TOP498, Ruminococcaceae UCG-005, Blautia, Ruminiclostridium 9 and Clostridium phoceensis) were modified. Crosstalk between GM and systemic tryptophan metabolism.
Zhong et al., 2021 [30]	Randomized controlled trial	*n* = 12 (F) previously inactive older healthy subjects, exercise (*n* = 6) and control (*n* = 6)	Exercise: 69.83 (±4.50) Control: 67.50 (±4.28)	PE: aerobic and anaerobic	1 h 4 × wk combined aerobic and resistance exercises (progressive overload)	Not controlled or recorded	8 weeks	16S rRNA GA V4F/V4R region	No changes in α-diversity. ↑ Prevotella, ↑ Verrucomicrobia, ↓ Proteobacteria abundance in the exercise group.
Moitinho-Silva et al., 2021 [31]	Randomized controlled trial	*n* = 36 (M/F) healthy physical inactive subjects, endurance (*n* = 12) and strength exercises (*n* = 13) with control (*n* = 11). Elite athletes for comparison (*n* = 13)	Endurance: 31.4 (±8.3) Strength: 29.9 (±7.9) Control: 33.4 (±7.9) Elite: 30 (±9.9)	PE:aerobic or anaerobic	Endurance: 30′ (at least) 3 × wk running Strength: 30′ 3 × wk whole-body hypertrophy strength training	Food questionnaire Elite: no data. Macronutrients, fiber, and total energy intake.	6 weeks	16S rRNA GA V1/V2 region	No specific bacteria changes. GM change patterns largely varied among individuals of the same group. No differences in α-diversity between elite and physical inactive subjects.

M: male; F: female; BMI: body mass index; ↑: increase; ↓: decrease; GA: gene amplification; FFQ: food frequency questionnaire; CRF: cardiorespiratory fitness; VO_2_peak: peak oxygen uptake; GM: gut microbiome; SCFAs: short-chain fatty acids; wk: week/s; PA: physical activity; PE: physical exercise; IPAQ: international physical activity questionnaire; WHO: world health organization; CI: confidence interval; VO_2max_: maximal oxygen uptake; HHR: heart rate reserve; ce: cycle ergometer; LTPA: leisure-time physical activity; METs: metabolic equivalent of task; HIIT: high-intensity interval training; T2D: type 2 diabetes; HR_max_: maximal heart rate; rpm: revolutions per minute; PUFA: polyunsaturated fatty acids.

## Data Availability

The data presented in this study was obtained from the included studies and was openly available.

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
