# Peer review of "The Relationship between Physical Activity, Physical Exercise, and Human Gut Microbiota in Healthy and Unhealthy Subjects: A Systematic Review"

_biology, 2022, doi:10.3390/biology11030479_

Round 1

Reviewer 1 Report

I think that the work performed is impressive and very extensive. I would just suggest to extend the methodological explanation in section 2.5. Risk of Bias within Studies.

Supplementary comments:

What are the main claims of the paper and how significant are they?

The authors claim that physical activity/exercise (PA/PE) could modify the composition of the gut microbiota, based on the meta-analysis of 25 full-text articles selected after careful curation. The authors also observed that PA/PE seems to counteract gut microbiota alterations related to obesity and type 2 diabetes, probably by reducing inflammatory signals, and that athletes may have enhanced beneficial gut microbiota functions. The fact that the authors made such a thorough research in the form of meta-analysis is significant to the field, since these questions are commonly considered but few articles studied it in a proper manner in humans.
How does the paper stand out from others in its field? The methodology is very detailed and the Results are not biased.
Are the claims novel? If not, which published papers compromise novelty? There is this meta-analysis: https://pubmed.ncbi.nlm.nih.gov/33630874/ I would like the authors to state why their work is different from this one.
Are the claims convincing? If not, what further evidence is needed? They are convincing, but I just said that the bias section should be extended.
Are there other experiments or work that would strengthen the paper further? I don't think that it is necessary.
How much would further work improve it, and how difficult would this be? Would it take a long time? I don't think that it is necessary.   Are the claims appropriately discussed in the context of previous literature? Yes

Reviewer 2 Report

Please find the attached document.

Reviewer 3 Report

The article of Cataldi, S. et al. reviewed the scientific literature about the relationship between physical activity or exercise and human gut microbiota in healthy and unhealthy subjects. The study is mostly complete. The conclusions drawn are well supported by the literature analysis. The authors have also discussed the limitation of the theme reviewed. The quality of the presentation is good, in accordance with the way and stages of how the articles reviewed were analyzed. I support the publication of the manuscript.

Author Response

Dear reviewer,

thank you very much for your comments!